# A Portable and Wireless Multi-Channel Acquisition System for Physiological Signal Measurements [note 1]

**DOI:** 10.3390/s19235314

**Published:** 2019-12-03

**Authors:** Shing-Hong Liu, Jia-Jung Wang, Tan-Hsu Tan

**Affiliations:** 1Department of Computer Science and Information Engineering, Chaoyang University of Technology, Taichung 413, Taiwan; shliu@cyut.edu.tw; 2Department of Biomedical Engineering, I-Shou University, Kaohsiung 824, Taiwan; 3Department of Electrical Engineering, National Taipei University of Technology, Taipei 106, Taiwan; thtan@mail.ntut.edu.tw

**Keywords:** acquisition system, physiological signal, portable device, application program

## Abstract

We propose a portable and wireless acquisition system to help consumers or users register important physiological signals. The acquisition system mainly consists of a portable device, a graphic user interface (GUI), and an application program for displaying the signals on a notebook (NB) computer or a smart device. Essential characteristics of the portable device include eight measuring channels, a powerful microcontroller unit, a lithium battery, Bluetooth 3.0 data transmission, and a built-in 2 GB flash memory. In addition, the signals that are measured can be displayed on a tablet, a smart phone, or a notebook computer concurrently. Additionally, the proposed system provides extra power supply sources of ±3 V for the usage of external circuits. On the other hand, consumers or users can design their own sensing circuits and combine them with this system to carry out ubiquitous physiological studies. Four major advantages in the proposed system are the capability of combining it with a NB computer or a smart phone to display the signals being measured in real time, the superior mobility due to its own independent power system, flash memory, and good expandability. Briefly, this acquisition system offers consumers or users a convenient and portable studying tool to measure dynamic vital signals of interest in psychological and physiological research fields.

## 1. Introduction

Most fitness and healthcare devices are capable of dynamic monitoring; for instance pedometers and heart rate monitors. These wearable devices must have the function of an independent power supply and the ability of wireless transmission. Typically, they measure and deal with these physiological signals by means of a microcontroller unit (MCU) or send these signals to a tablet or a smart phone to be further processed. Therefore, in the development of wearable healthcare devices, it is important to explore innovative measurement technologies and signal processing methods to overcome the instability relevant to the acquisition of dynamic physiological signals. For these researchers in hardware development, they must not only develop novel analog driving circuits, but also build a standard acquisition system which is supposed to incorporate a power system, an analog-to-digital converter, and a wireless transmission scheme. Moreover, they also need to build a display and control interface in a software environment. It is a really time and energy-consuming task for those researchers in their ordinary studying procedure. Thus, how to efficiently shorten their research time may become a serious issue. 

Multi-channel physiological signal measurement systems that are employed in a lot of physiological experiments usually require alternating current (AC) and must connect the different sensor modules with the lead wires. For example, both the BioPac MP150 system and the K&H KL-710 system can connect with specific physiological measurement modules and then record and display multiple physiological signals on a notebook (NB) or personal computer [1,2]. Obviously, several researchers make use of compact DAQ cards to assemble their own signal acquisition systems, which connect with NB computers through a universal serial bus (USB) cable. However, the dimensions of these acquisition systems would be huge, and their power sources are supported by AC or by a built-in battery in the NB computer. Of these two acquisition systems, the BioPac MP150 system and the K&H KL-710 system, neither have an independent data storage unit, nor a wireless transmission function. Moreover, these two acquisition systems are not able to be operated by an application program (APP) using a tablet or a smart phone. Due to those serious limitations, it is difficult for researchers to apply these systems to developing their own wearable devices. In addition, the BioPac Bionomadix smart center can connect three wireless dual-channel modules [3], and the Bioradio^TM^, with eight channels, is a wireless measurement system which can concurrently measure six different physiological signals [4]. 

Since Bluetooth (BT) technology was not prevalent in the iOS and the Android operating systems ten years ago, and the performance of the central processing unit (CPU) in mobile phones was relatively low, a mobile phone did not be used to concurrently display and deal with these measurements in real time. The BT3.0 and BT4.0 dual-channel modules both have become the standard components of smart phones in the past few years, and the CPU performance in modern mobile phones has been considerably upgraded. Therefore, it is now feasible to develop a portable wireless system based on the traditional NB computer or smart phone that can measure, store, and display multiple physiological signals in real time. This acquired system can really help researchers to develop a wearable device to apply in the fitness and healthcare fields.

In previous studies, Lin et al. [5] adopted the MSP430 microcontroller as the core of an Internet of Things (IoT)-based, wireless, intelligent polysomnography system. Their system can transmit the measured signals directly to a mobile phone for storage and analysis. Because the problem of the slow BT transmission speed, the sampling frequency of the system had to be reduced to 125 Hz, and the system lacked the function to access data. Some studies used the wireless technique, the IEEE 802.15.4 standard, to transmit physiological signals to a personal computer. Biagetti et al. built an acquisition system to measure the electrocardiogram (ECG) and electromyogram (EMG) signals by applying a radio frequency (RF) transceiver for wireless communication [6]. Of course, each wireless note also needs the power to transmit data to the server station. Bhutta et al. used the WiFi technique to transmit data [7]. Dey et al. utilized the Zigbee technology to establish a wireless sensor network for ECG measurements [8]. Apparently, all these modules did not connect directly with smart devices, such as a tablet or a smart phone. Therefore, such studies may be limited to doing the experiments under ubiquitous situations. Liu et al. developed a wearable device that could measure ECG and gestural movement signals [9], but this device could not demonstrate the signals being recorded in real time. On the other hand, the data had to be downloaded offline from the device to display the signals’ measurements. Hsu et al. measured the galvanic skin response with electrical impedance spectroscopy, which used the BT technology to transmit data to smart phones [10]. Milici et al. designed a thermo sensor to detect the changing in airflow during breathing [11]. They also used the BT technique to transmit the acquired data to a smart phone for long-term recording. Meanwhile, Liu et al. developed a preliminary prototype of a portable multi-channel physiological measurement system [12].

Since many sensors and their driver circuits have been integrated as chips, their power consumption has been reduced and their size has been miniaturized. For example, the Analog Device AD8232 is an analogy integrated chip for the ECG measurement [13], and the MAX30101 of Maxim Integrated is a digital chip for the oxyhemoglobin saturation assessment [14]. The ADXL325 of the Analog Device is an analog tri-axial accelerometer integrated chip applied for object activity measurement [15]. As the sensor modules are employed to detect a variety of physiological signals, an acquisition system with multiple channels is required to show and record these signals. However, among the current commercial products, merely a few devices can not only be controlled with a tablet or a smart phone, but can also store data on them. Thus, the goal of this study was to develop a portable and wireless multi-channel acquisition system for the physiological signal measurements. It has eight analog channels and can be controlled by a NB computer or a smart device. The measured signals can either be shown on a NB computer or a smart device in real time, or be stored on the flash memory of the portable acquisition device. A sampling frequency of the portable signal acquisition device is 500 Hz, which is enough to conform to the Nyquist frequency of some physiological signals, such as the ECG, electroencephalogram (EEG), electrooculogram (EOG), galvanic skin response (GSR), and photoplethysmogram (PPG), since most of them do not have a large bandwidth [16]. With a TI MSP430 F5438A as its MCU, the portable acquisition device has a compact size, uses a lithium battery (350 mA) to supply the needed power, employs a BT3.0 module to transmit data, and a 2 GB flash memory to store the signals being measured. Moreover, the portable acquisition device can offer dual power levels, ±3 voltage, so that the external sensor modules may connect with this device to measure different physiological signals. The real-time measurements can be displayed on a NB computer or smart phone. Thus, consumers or researchers can confirm the stability and accuracy of the measured signals during the experiment. 

The rest of the paper is organized as follows: Section 2 describes the structure of the multi-channel acquisition system, and its software commands on both smart devices and NB computers. Section 3 describes the hardware and firmware designs of the portable signal acquisition device. Section 4 presents the results, and the discussion. Conclusions are drawn in Section 5 and Section 6.

## 2. Measurement System

The structural diagram of the portable and wireless multi-channel acquisition system, which includes three parts, a portable acquisition device, a graphic user interface (GUI), and an application program (APP) for a NB computer and a smart device, is shown in Figure 1. There are eight analog channels available to connect with the external sensor modules indicated with an orange arrow. The black arrows represent the control commands which are sent out by a smart device or a NB computer. The control commands can trigger the portable acquisition device to either start or stop the measurement, write the data to the flash memory, or clear the data on the flash memory. The data on the flash memory of the portable acquisition device is downloaded to the NB computer by means of the BT or a universal serial bus (USB). Then, the downloaded data can be displayed on the NB computer. The blue arrows represent the measured physiological signals that can be shown on the smart device or the NB computer in real time. When the measured signals are transmitted from the portable acquisition device to the NB computer through either the BT3.0 or the USB; the sampling rate is 500 Hz. When the portable acquisition device transmits one signal to the smart device, the sampling rate is reduced to 100 Hz. The software commands on smart devices and NB computers are described in the subsections below.

### 2.1. Application Program

In this study, Cordova and Eclipse were chosen to develop an APP for smart devices, such as a tablet or a smart phone. As we know, Cordova is an open source software and many users can employ it to discuss how to create new APPs online. Eclipse is also a well-known cross-platform which belongs to a free integrated development environment, with lots of external programs. Thus, it can be utilized flexibly and conveniently.

We selected, in this work, JavaScript as the programming language to develop an APP. In addition, both the built-in BT module and BT plug-in of the Android system are employed to connect with the acquisition system and to send out commands, such as triggering the analog to digital converter (ADC), stopping the ADC, writing data to the flash memory, and finishing the writing of the data. As shown in Figure 2, the commands inside the red block belong to the APP.

### 2.2. Graphic User Interface

As shown in Figure 2, the NB GUI not only has the same commands as the APP, but also has the “Load” and “Clear” commands. The loading command is used to download the data stored on the flash memory to the NB computer, while the clear command is used to erase the data on the flash memory. There are four bytes in each command package. The first two bytes of the package are the head bytes, 0 × 66. Six commands are proposed in this acquisition system and their corresponding codes are shown in Table 1.

## 3. Hardware and Firmware of the Portable Signal Acquisition Device

In this section, the major parts of the portable signal acquisition device are described, respectively, including the power circuit, the communication protocol of the BT module, the storage arrangement of the flash memory, and the principal settings of the TI MSP430 F5438A.

### 3.1. Power Circuit

In the power circuit of the portable signal acquisition device, the Texas Instruments BQ24072 battery charger is employed. The interatged circuit (IC) (TPS78233, IT) is applied to provide a voltage of 3.3 V, which is supplied from a lithium battery, while the IC (TPS60400, IT) is used to provide a voltage of −3.3 V. This power scheme also offers the dual power, ±3.3 V, which can be utilized by external sensor circuits.

### 3.2. Communication Protocol of Bluetooth

The BT 3.0 module (BTM-204B) was chosen to connect with the Universal Asynchronous Receiver/Transmitter (UART) of the MSP430 F5438A. There are 18 bytes in the transmission package. First, the two bytes, 0 × 66, are used as the head bytes of the package data, and the ADC value for each channel is divided into high bytes and low bytes. The purpose of the two head bytes is to mark the start of the data entry. After the starting information is received, the deciphering operation includes the multiplication of the high bytes of ADC value by 0 × 100. Then, this value is added to the low bytes as the true value. The UART port of the MSP430 F5438A is selected as a virtual USB port by the PL2303HXD IC for communicating with the NB computer.

### 3.3. Storage Format of Flash Memory

Since the built-in 256 kb flash memory of the MSP430 is insufficient to record a large amount of data, external add-on memory is required for the portable signal acquisition device. A NAND flash memory (GD5F2Q4UBFI) of 2 GB is employed in the proposed system. The array of the flash memory consists of 64 pages in total. The first 2 bytes of the first page specify the file number, which is followed sequentially by the start page, the last page, and the last bytes. There are 4096 bytes in total, for the first page. Since the file name occupies the first 2 bytes, six bytes are left. Therefore, the 2 GB flash memory can be used to store 682 files. Figure 3 shows the structure of one data access file in the 682 files. NAND Flash uses the SPI (serial peripheral interface) communication approach for data transfers to the MSP430 microcontroller, in which the clock (CLK) frequency is 12 MHz and the data transmission speed is nine kilobits per second at a sampling frequency of 500 Hz. There are eight channels, each channel signal being assigned 2 bytes, with a transmission rate of 8 kilobits per second. Therefore, the access rate of the portable signal acquisition device is good enough.

### 3.4. The Settings of MSP430F5438A

The MSP430 F5438A, a 16-bit MCU used in this study has one flash memory of 256 kilobytes and eight ADCs of 12 bits each. Each ADC channel with a sampling frequency of 500 HZ and an internal flash memory is used to record the number and the size of a file accessed in the external flash memory. The firmware flowchart for the MSP430 microcontroller is shown in Figure 4. There are two UART sets in the 16-bit MCU. One is used for the BT module and the other for the USB port. When the measurement signals are sent to a smart device to determine whether they are saved to the flash memory or not, the sampling frequency is reduced to 100 Hz. The main clock rate of the MCU is set at 24 MHz. Clock A0, at 500 Hz, is employed as the ADC sampling frequency and to set up the UART and SPI clocks. The eight channels are made available for the ADC operation. After initial settings are completed, time interruption is enabled and the MCU will wait for a command, such as an “ADC on,” a “Write data” to flash memory, and so on.

The MCU determines whether the flash memory has enough free space based on two conditions: (1) there are 682 files in the flash memory, and (2) the last page of the flash memory has been used. When either of these two conditions is satisfied, the data can no longer be written to the flash memory.

The MCU controls a set of four LEDs indicating: (1) the status of the connection to the USB port, (2) the status of the BT connection, (3) the status of the 3.3 V power, and (4) the status of the flash memory, respectively. If the MCU is in the reading mode, the LED will remain on. However, when in the writing mode, a blinking LED is displayed.

## 4. Results

In this section, the circuit structure and the electric performance of the portable signal acquisition device are depicted. The GUIs on the notebook and smart phone designed to operate the function of the portable signal acquisition device and the display of measured signals are also presented.

### 4.1. Circuit of Portable Signal Acquisition Device

Figure 5a shows the block diagram of the portable acquisition device, Figure 5b shows its printed circuit, and Figure 5c shows its real photograph. Without the battery, its weight is only 15 g, and its size is 12 × 12 cm. In the upper right-hand corner, two voltage sources, 3.3 and −3.3 V, provide the power for external sensor devices with two maximum currents of 400 and 60 mA, respectively. The USB port is used either to charge the lithium battery or to download the data to the hard disk in the NB computer. In Figure 5a, the black lines represent the power connection, and the blue lines represent the data transmission. The 5 V line and ground line of the USB port were connected to the input pins of the charge IC, TI BQ24072. The two data lines of the USB port were connected to the USB-to-serial bridge controller, PL 2303HX. The USB socket is placed at the upper left- hand corner of the printed circuit board. The MCU, BT module, and flash memory consume the largest amount of power in this portable acquisition device. Since the active current of the MSP430 F5438A is 165 μA/MHz, it requires the currents of 3.96 mA and 2.6 μA under the maximum running conditions and the low power mode, respectively. The BT model needs the currents of 37 mA and 70 μA under the transmission and standby modes, respectively. Additionally, the flash memory needs the currents of 40 mA and 70 μA under the operation and standby modes, respectively. Therefore, when the portable acquisition device makes use of a smart device to display the signals, it requires a maximum current of about 43 mA. But, if users want to display signals and write data to the flash memory at the same time, the maximum current conumed will be 73 mA. When the input of ADC is connected to the power source, +3.3 V, and ground, 0 V, the codes of ADC with the ten samples are shown in Table 2. The statistical codes for the positive power source and ground source are 4095 ± 1 (mean ± standard deviation) and 1 ± 1, respectively.

### 4.2. Notebook Computer GUI

As the communication interface software, GUI codes for the NB computer were written in C#. The main function of the NB computer is the system control and downloading data from the portable signal acquisition device. The GUI software was developed for the connection with the portable signal acquisition device, and for the control operations, including starting the ADC, stopping the ADC, writing data to the flash memory, loading and clearing the data of the flash memory, and triggering the other actions.

The portable acquisition devices connected with three sensor modules include the KL-74002A (ECG), KL-74006A (PPG), and KL-74005 (heart sound). These sensor modules were commercial products designed by the K&H Co., Ltd. Company, Taiwan. In the GUI, the first operation is to select one of the common ports (COM PORT) to link the portable signal acquisition device, as shown in Figure 6. Once this connection has been made, the “Connect” button on the display is indented. If the portable signal acquisition device and the NB computer have been successfully linked, the Connect button will change to Disconnect, and the other commands such as the ADC ON, Load Flash, and Clear Flash will appear, and any one of them can be used. After the ADC ON button has been pressed, the measurement signals will be immediately sent to the NB computer. Then, users can choose which channels will be used to display the signals, as shown in Figure 6. There are three kinds of measurement signals respectively, from the channels 1, 2, and 4 that can be displayed on the screen—ECG, PPG, and heart sound signals. To stop the ADC operation, press the ADC OFF button. If users want to record the signals, they can press the Write button. These signals will be recorded on the flash memory in real time. For each new recording event, the file number in the flash memory is automatically increased by one, and the data will be recorded on a new page of the flash memory.

If users need to download the data from the flash memory to the NB computer, they may press the Load Flash button. An access window will be displayed immediately. Select a location and enter a file name first; then, press the Save button to allow all the data on the flash memory to be saved to a file in a “text” format, and the data of each experiment can be recorded in an independent file. The information associated with the saved files will be displayed when the data on the flash memory has been successfully saved, as shown in Figure 7. Such file information includes the file numbers, the downloading bytes, the ratio of the flash memory used, and the downloading condition. The data of the eight channels in the first file are shown in the left column of Figure 7.

### 4.3. Application Program

Figure 8 illustrates the APP interface where the numbers with the red color, are used to indicate the processing steps. First, as BT Search is initiated, the corresponding BT device can be found by name. After establishing a successful connection, users may then press the ADC_ ON button to begin the signal measurement. But, only the signal of the first channel will be displayed in the APP, as shown in Figure 6. When the Flash_Write button is pressed, the signals of the eight channels are recorded on the flash memory. The Flash_Stop button is used to stop the recording. Finally, users may press the ADC_Off button to turn the ADC operation off.

## 5. Discussion

In the last decade, wearable devices have been applied in various fields, especially in the healthcare sector [17]. Among the physiological measurements, the ECG, EMG, EEG, and PPG signals are the most popular in modern wearable devices for daily healthcare monitoring. However, one of the challenges with designing such wearable devices is how to faithfully acquire these physiological signals under an actual environment. Requirements for the wearable healthcare devices are somewhat different from those for the traditional devices in clinical locations. A big difference is that a wearable device always not only detects the dynamic physiological signals, but needs to display these signals in real time. In recent studies, some acquisition systems have employed BT, Zigbee, WiFi, or RF transceiver to transmit the measurement signals to a NB computer or a smart device, and to display or store them on those smart devices [5,6,7,8]. The characteristic comparison among the four wireless techniques is shown in Table 3. Although the WiFi has the fastest transmission rate and the longest transmission distance, its power consumption also is the largest. The BT does not have the best performance of these wireless techniques. But, nowadays, both NB computers and smart phones have WiFi and BT communication functions. Nevertheless, there are two main disadvantages in those acquisition systems, including low mobility and low sampling rates.

In this study, a portable and wireless multi-channel acquisition system for physiological signal measurements was fully established. This acquisition system is primarily comprised of a portable acquisition device, a GUI, and an APP to display the signals on a NB computer or a smart device. Since the power consumption of the BT technique is relatively low compared with WiFi, we finally chose the BT technique to perform the connection with the NB computer or the smart device in order to reduce the total power consumption of the portable acquisition device. Three major advantages exist in the present multi-channel acquisition system. First, this acquisition system allows users to combine it with a NB computer or a smart device to display the measurement signals in real time and to easily control the functions of the portable acquisition device. Therefore, users are permitted to check the stability and accuracy of the recorded signals in real time during the experiment. When the correctness of the measured signals is carefully confirmed, users can begin to record the signals on the flash memory. Second, the portable signal acquisition device possesses superior mobility. Users may take the device to any place due to its own independent power system and adequate memory. Third, the acquisition system has good expandability. For instance, it offers ±3.3 voltage for the use of external or additional circuits. Furthermore, with eight channels, it can be applied to simultaneously register various physiological signals. Because the resolution of ADC is 80.6 uV, it could be used to properly measure several kinds of physiological signals—ECG, EMG, PPG, body acceleration signal, and so on. 

The commercial products, such as the BioPac Bionomadix smart center [3] and Bioradio^TM^ wireless measurement system [4], have the exclusive sensor modules and must connect with the NB. These measurement systems are just like gateways to transmit the signals to the NB computer. Therefore, their compatibilities are not higher than our proposed system. Users need to carry a NB computer to record these signals during the experimental course. For our system, users only use a smart phone to check the measurement signals and to see whether they are stable and accurate or not when doing the experiment. Once the signals are properly measured, users can control the portable acquisition device to record the signals onto the flash memory in real time. Moreover, since our proposed system is intended to be employed for signal acquisition, it does not need to do the calibration of the sensor modules.

## 6. Conclusions

In summary, we designed the portable and wireless multi-channel acquisition system which offered a convenient and portable studying tool to measure the dynamic physiological signals in healthcare research fields. Users can watch the measurement signals in real time and control the functions of this multi-channel acquisition device with a smart phone or NB computer. Additionally, researchers can design their own sensing circuits and combine them with this acquisition system. Then, they can rapidly perform their experiments, and with a clear mind, collect physiological signals of interest in any scenario. Our proposed acquisition system is currently a prototype. However, in the future, we will modify this system as a commercial product according to the feedback of the intended audiences.

## Figures and Tables

**Figure 1 sensors-19-05314-f001:**
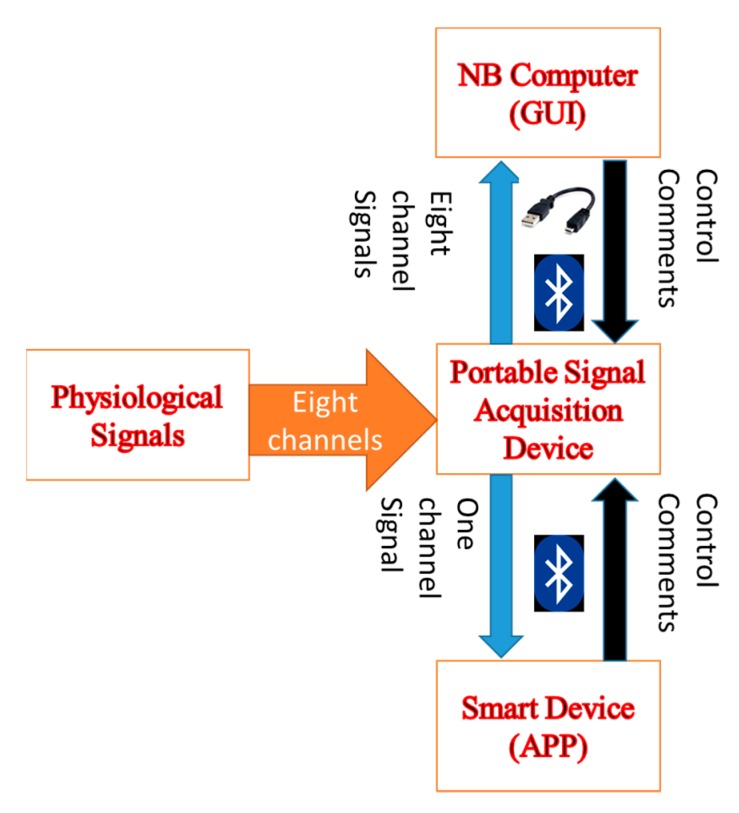
Framework of the portable and wireless multi-channel acquisition system.

**Figure 2 sensors-19-05314-f002:**
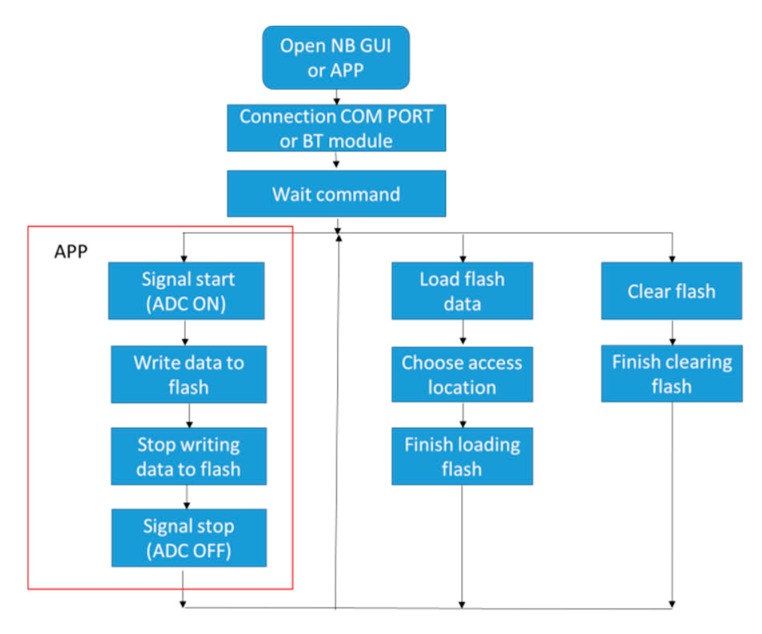
Command flowchart of the APP and notebook (NB) computer.

**Figure 3 sensors-19-05314-f003:**
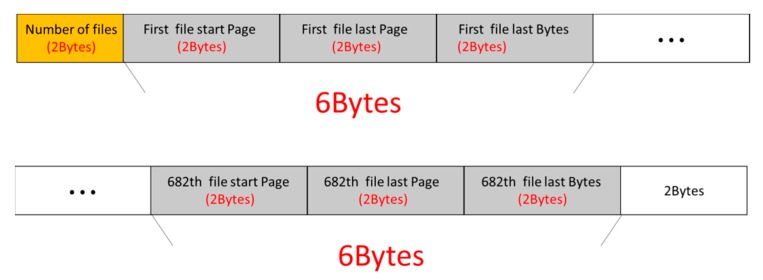
Flash file storage structure.

**Figure 4 sensors-19-05314-f004:**
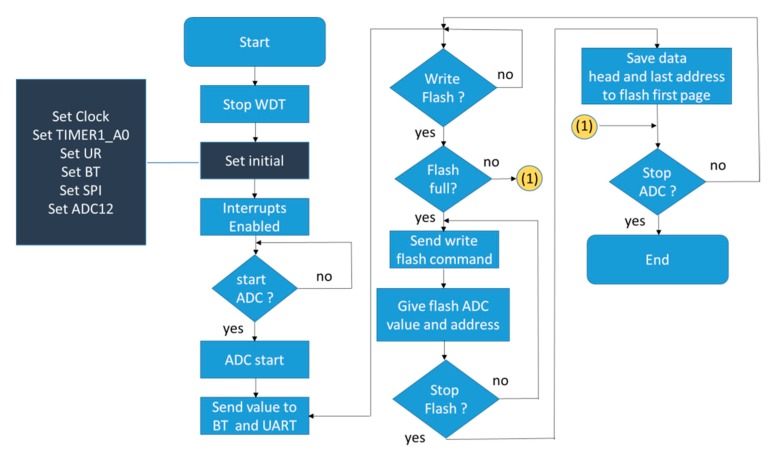
MSP430 firmware flowchart.

**Figure 5 sensors-19-05314-f005:**
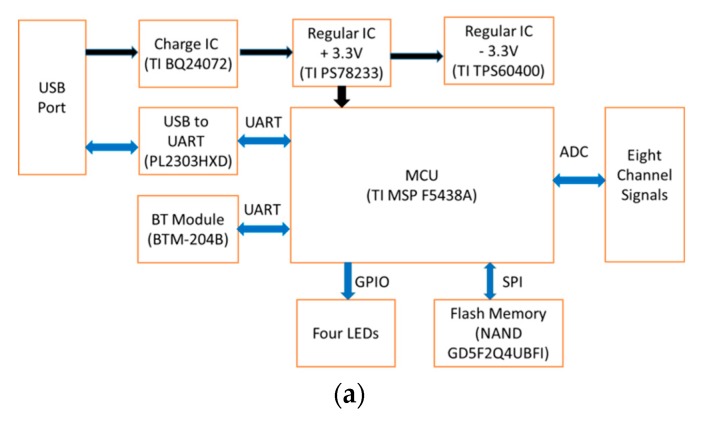
The portable acquisition device, (**a**) its block diagram, (**b**) its printed circuit, and (**c**) a real photograph of it.

**Figure 6 sensors-19-05314-f006:**
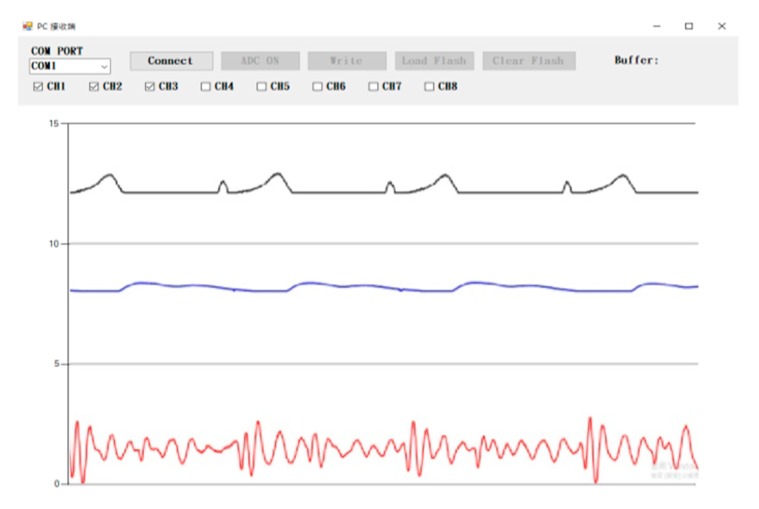
Three signals are displayed on the graphic user interface (GUI) of the NB computer.

**Figure 7 sensors-19-05314-f007:**
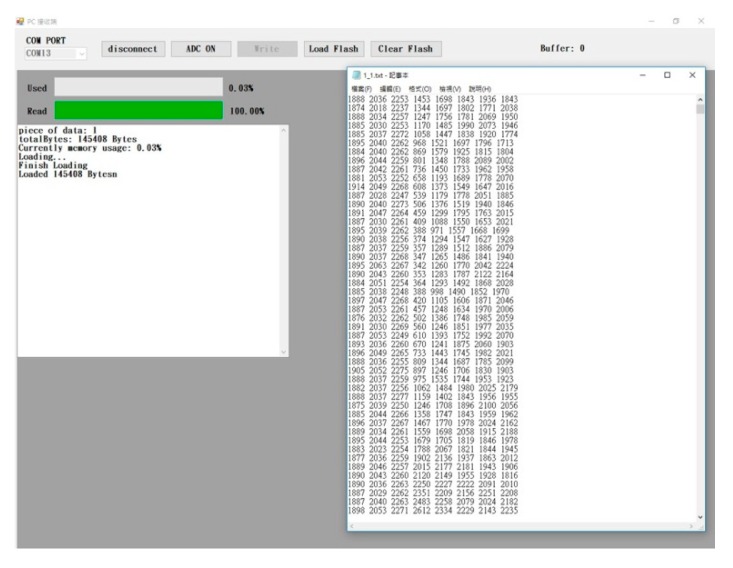
The Load flash memory interface and txt file contents.

**Figure 8 sensors-19-05314-f008:**
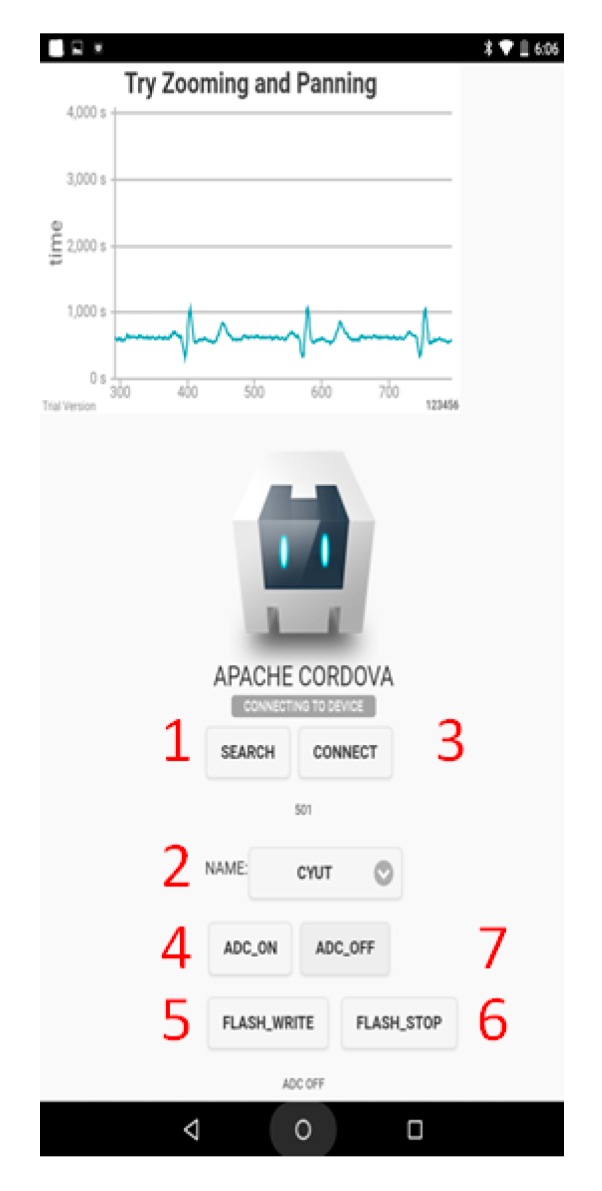
Display screen of the APP.

**Table 1 sensors-19-05314-t001:** The codes of the control commands.

Code	Commands
0X3031	ADC ON
0X3032	ADC OFF
0X3033	Load flash
0X3034	Clear flash
0X3035	Write data to flash
0X3036	Stop writing data to flash

**Table 2 sensors-19-05314-t002:** The statistical error of analogue to digital converter (ADC) codes under +3.3 V and 0 V.

Source	1	2	3	4	5	6	7	8	9	10	Mean ± SD ^1^
+3.3 V	4095	4095	4095	4093	4094	4095	4095	4095	4094	4095	4094.5 ± 0.70
0 V	1	1	0	1	2	0	0	0	1	0	0.6 ± 0.70

^1^ SD: standard deviation.

**Table 3 sensors-19-05314-t003:** The characteristic comparison among the four wireless techniques.

Technique	Distance	Power Consumption	Transmission Rate
BT 3.0 (HC-05) [18]	<10 m	30 mA	1M (bits/s)
WiFi (ESP8285) [19]	>10 m	80 mA	54M (bits/s)
Zigbee (Xbee^®^) [20]	>10 m	45 mA	250k (bits/s)
RF (BC9824) [21]	<2 m	10 mA	2M (bits/s)

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
