# Peer review of "A Portable and Wireless Multi-Channel Acquisition System for Physiological Signal Measurementsâ€"

_sensors, 2019, doi:10.3390/s19235314_

Round 1
Reviewer 1 Report
Comments:
This manuscript discusses about the development of an eight-channel wireless portable physiological signal acquisition system. Along with the portable system the authors have developed and GUI for the notebook (NB) and an application program (APP) for the mobile devices. In comparison to the previous studies, the authors have claimed that their proposed system has the capability of portability, because of its built-in battery unit, and also the acquired signals can be displayed on the NB and mobile device along with storing the data on the onboard flash memory. However, there are several uncertainties that should be clarified.
Major Issues:
The introduction part of the manuscript is incomplete. The authors have not completely reviewed the previous systems which are already been developed by other researchers to detect the physiological signals from human body. In the proposed system, the authors have used Bluetooth technology to transmit the data to mobile devices. The reviewer has an opinion that rather than Bluetooth, WiFi, technology is more suitable for the health care applications because of its multiple benefits over the Bluetooth technology like: i) speed of data transmission, ii) line of sight problem in Bluetooth, and etc.- Bhutta, M. R., Hong, K.-S., Kim, B.-M., Hong, M. J., Kim, Y.-H., and S.-H., (2014). Note: Three wavelengths near-infrared spectroscopy system for compensating the light absorbance by water. Review of Scientific Instruments. Vol. 85, No. 2, AN: 026111, pp1-3.
The authors should include a true pictures of the developed system. The authors have shown some signals outputs in figure 4 and 6 but they have not mentioned in the manuscript that which sensor they have attached with the input to acquire the respective graphs. The authors have not mentioned in the manuscript that their system is capable of which physiological sensors to acquire the data. In the proposed system, the authors are using the USB port for charging the battery and also for data movement to the NB (Figure 3 (a)). But authors have not mentioned that whether they have multiple USB ports in the system or they have used a single USB port with the time multiplexing capability. In the proposed system, the authors are sending the data to the NB by default to display on the GUI then why are they using the onboard flash memory whereas they can easily store the data in the NB. In the manuscript the authors have not verified the acquired signals using the statistical data.- Hong, K.-S., Bhutta, M. R., Liu, X., and Shin, Y.-I. (2017). Classification of somatosensory cortex activities using fNIRS. Behavioural Brain Research, 333, 225-234.
- Hong, K.-S., and Santosa, H. (2016). Decoding four different sound-categories in the auditory cortex using functional near-infrared spectroscopy. Hearing Research, 333, 157-166.
- Bhutta, M. R., Hong, M. J., Kim, Y.-H., & Hong, K.-S. (2015). Single-trial lie detection using a combined fNIRS-polygraph system. Frontiers in Psychology, 6, 709.
In this manuscript the authors have not made any discussion about the signal o Noise Ratio for their developed system. The authors have mentioned about any precautionary or security measure for the communication errors or data loss while transmitting data to and from the developed system. The authors have not mentioned any LED numbers for the specified jobs which they have mentioned in the line numbers 201 and 202. The Discussion section of the manuscript is only a small copy of the Introduction section. In Discussion section the authors should discuss about the results they have obtained, any flaws which they have in system and/or the future of the developed system.Minor issues:
There are some spelling and indentation errors in the manuscript, which need to be corrected.Author Response
To Reviewer #1:
Thank the first reviewer for his/her valuable comments that make better this manuscript. The texts in this revised manuscript have been corrected/ modified by red words.
Comments to the Author
This manuscript discusses about the development of an eight-channel wireless portable physiological signal acquisition system. Along with the portable system the authors have developed and GUI for the notebook (NB) and an application program (APP) for the mobile devices. In comparison to the previous studies, the authors have claimed that their proposed system has the capability of portability, because of its built-in battery unit, and also the acquired signals can be displayed on the NB and mobile device along with storing the data on the onboard flash memory. However, there are several uncertainties that should be clarified.
1) The introduction part of the manuscript is incomplete. The authors have not completely reviewed the previous systems which are already been developed by other researchers to detect the physiological signals from human body.
Ans: The authors have modified the Introduction section and reviewed some commercial products which are be similar to our proposed system.
IntroductionMost of the fitness and healthcare devices are capable of dynamic monitoring, such as pedometers and heart rate monitors. These wearable devices must have the function of an independent power supply and the ability of wireless transmission. Typically, they measure and deal with these physiological signals by means of a microcontroller unit (MCU) or send these signals to a tablet or a smart phone to further process. Therefore, in the development of wearable healthcare devices, it is important to explore innovative measurement technologies and signal processing methods to overcome the instability relevant to the acquisition of dynamic physiological signals. For these researchers in the hardware development, they must not only develop novel analog driving circuits, but also build a standard acquiring system which is supposed to incorporate a power system, an analog-to-digital converter and a wireless transmission scheme. Moreover, they also need to build a display and control interface in the software development. It is a really time- and energy-consuming task for those researchers in their ordinary studying procedure. Thus, how to efficiently shorten their research time may become a serious issue.
Multi-channel physiological signal measurement systems that are employed in a lot of physiological experiments usually require an alternating current (AC) and must connect the different sensor modules with the lead wires. For example, both the BioPac MP150 system and the K&H KL-710 system can connect with specific physiological measurement modules and then record and display multiple physiological signals on a notebook (NB) or personal computer [1,2]. Obviously, several researchers make use of compact DAQ cards to assemble their own signal acquisition systems, which connect with NB computers through a universal serial bus (USB) cable. However, the dimension of these acquisition systems would be huge, and their powers are supported by an AC or by a built-in battery in the NB computer. These two acquisition systems, the BioPac MP150 system and the K&H KL-710 system, neither have an independent data storage unit, nor have a wireless transmission function. Moreover, these two acquisition systems are not able to be operated by an application program (APP) using a tablet or a smart phone. Due to those serious limitations, it is difficult for researchers to apply these systems in developing their own wearable devices. In addition, the BioPac Bionomadix smart center can connect three wireless dual-channel modules [3], and the BioradioTM, with eight channels, is a wireless measurement system which can concurrently measure six different physiological signals [4].
Since the Bluetooth (BT) technology was not prevalent in the iOS and the Android operating systems ten years ago, and the performance of the central processing unit (CPU) in mobile phones was relatively low, a mobile phone did not be used to concurrently display and deal with these measured signals in real time. The BT3.0 and BT4.0 dual-channel modules both have become the standard components of smart phones for the past few years, and the CPU performance in modern mobile phones has been considerably upgraded. Therefore, it is now feasible to develop a portable wireless system based on the traditional NB computer or smart phone that can measure, store and exhibit multiple physiological signals in real time. This acquired system can really help researchers to develop a wearable device to apply in the fitness and healthcare fields.
In previous studies, Lin et al. [5] adopted the MSP430 microcontroller as the core of the IoT based wireless polysomnography intelligent system. Their system can transmit the measured signals directly to a mobile phone for storage and analysis. Because the problem of the slow BT transmission speed, the sampling frequency of the system had to be reduced to 125 Hz, and the system lacked the function to access data. Some studies used the wireless technique, the IEEE 802.15.4 standard, to transmit physiological signals to a personal computer. Biagetti et al. built an acquisition system to measure the electrocardiogram (ECG) and electromyogram (EMG) signals by applying a RF transceiver for wireless communication [6]. Of course, each wireless note also needs the power to transmit data to the server station. Bhutta et al. used the WiFi technique to transmit data [7]. Dey et al. utilized the Zigbee technology to establish a wireless sensor network for the ECG measurements [8]. Apparently, all these studies do not connect directly with smart devices, such as a tablet or a smart phone. Therefore, these studies may be limited to doing the experiments under ubiquitous situations. Liu et al. developed a wearable device that could measure the ECG and gestural movement signals [9], but this device could not demonstrate the recorded signals in real time. On the other hand, the data had to be offline downloaded from the device to display the measured signals. Hsu et al. measured the galvanic skin response with the electrical impedance spectroscopy, which used the BT technology to transmit data to smart phones [10]. Milici et al. designed a thermo sensor to detect the changing in airflow during breathing [11]. They also used the BT technique to transmit the acquired data to a smart phone for long-term recording. Meanwhile, Liu et al. have developed a preliminary prototype of a portable multi-channel physiological measurement system [12].
Since many sensors and their driver circuits have been integrated as chips, their power consumption has been reduced and their size has been miniaturized. For example, the Analog Device AD8232 is an analogy integrated chip for the ECG measurement [13], while the MAX30101 of Maxim Integrated is a digital chip for the oxyhemoglobin saturation assessment [14]. The ADXL325 of the Analog Device is an analog tri-axial accelerometer integrated chip applied for the object activity measurement [15]. As the sensor modules are employed to detect a variety of physiological signals, an acquisition system with multiple channels is required to show and record these signals. However, among the current commercial products, merely a few devices not only can be controlled with a tablet or a smart phone, but also can store data on them. Thus, the goal of this study is to develop a portable and wireless multi-channel acquisition system for the physiological signal measurements. It has eight analog channels and can be controlled by a NB computer or a smart device. The measured signals can either be shown on a NB computer or a smart device in real time, or be stored on the flash memory of the portable acquisition device. A sampling frequency of the portable signal acquisition device is 500 Hz which is enough to conform to the Nyquist frequency of some physiological signals, like as the ECG, electroencephalogram (EEG), electrooculogram (EOG), galvanic skin response (GSR), and photoplethysmogram (PPG), since most of them do not have a large bandwidth [16]. With a TI MSP430 F5438A as its MCU, the portable acquisition device has a compact size, uses a lithium battery (350 mA) to supply the needed power, employs a BT3.0 module to transmit data, and a 2GB flash memory to store the measured signals. Moreover, the portable acquisition device can offer dual power levels, ±3 voltage, so that the external sensor modules may connect with this device to measure different physiological signals. The real-time measuring signals can be displayed on a NB computer or smart phone. Thus, consumers or researchers can confirm the stability and accuracy of the measured signals during the experiment.
The rest of the paper is organized as follows: Section 2 describes the structure of the multi-channel acquisition system, and its software commands on both smart devices and NB computers. Section 3 describes the hardware and firmware designs of the portable signal acquisition device. Section 4 presents the results, and the discussion and conclusions are drawn in Section 5 and 6.
2) The reviewer has an opinion that rather than Bluetooth, WiFi, technology is more suitable for the health care applications because of its multiple benefits over the Bluetooth technology
Ans: We have added a reference paper related to the WiFi technology in the manuscript.
[7] Bhutta, M. R.; Hong, K.-S.; Kim, B.-M.; Hong, M.-J.; Kim, Y.-H.; Lee, S.H. Note: Three wavelengths near-infrared spectroscopy system for compensating the light absorbance by water. Review of Scientific Instruments. 2014, 85, 1-3.
3) The authors should include a true picture of the developed system.
Ans: We have added a real photograph of the portable acquisition device, as shown in Figure 5 (c).
4.1. Circuit of Portable Signal Acquisition Device
Figure 5(a) shows the block diagram of the portable acquisition device, Figure 5(b) shows its printed circuit, and Figure 5(c) shows its real photograph. Without the battery, its weight is only 15 g, and its size is 12 cm x 12 cm. In the upper right-hand corner, two voltage sources (3.3 V and -3.3 V) provide the power for external sensor devices with two maximum currents of 400 mA and 60 mA, respectively. The USB port is used either to charge the lithium battery or to download the data to the hard disk in the NB computer. The USB socket is placed at the upper left- hand corner of the printed circuit board. The MCU, BT module, and flash memory consume the largest amount of power in this portable acquisition device. Since the active current of the MSP430 F5438A is 165 μA/MHz, it requires the currents of 3.96 mA and 2.6 μA under the maximum running condition and the low power mode, respectively. The BT model needs the currents of 37 mA and 70 μA under the transmission and standby modes, respectively. Also, the flash memory needs the currents of 40 mA and 70 μA under the operation and standby modes, respectively. Therefore, when the portable acquisition device makes use of a smart device to display the signals, it requires a maximum current of about 43 mA. But, if users want to display signals and write data to the flash memory at the same time, the maximum consumed current will be 73 mA.
(a)
(b)
(c)
Figure 5. The portable acquisition device, (a) its block diagram, (b) its printed circuit, (c) its real photo.
4) The authors have shown some signals outputs in figure 4 and 6 but they have not mentioned in the manuscript that which sensor they have attached with the input to acquire the respective graphs. The authors have not mentioned in the manuscript that their system is capable of which physiological sensors to acquire the data.
Ans: We have described the sensor modules that can be used to detect signals in 4.2. Section.
4.2. Notebook Computer GUI
As the communication interface software, GUI codes for the NB computer are written in C#. The main function of the NB computer is the system control and downloading data from the portable signal acquisition device. The GUI software is developed for the connection with the portable signal acquisition device, and for the control operations, including starting the ADC, stopping the ADC, writing data to the flash memory, loading and clearing the data of the flash memory, and triggering the other actions.
The portable acquisition device connected with three sensor modules, including the KL-74002A (ECG), KL-74006A (PPG), and KL-74005 (heart sound). These sensor modules were commercial products designed by the K&H Co., Ltd. Company, Taiwan. In the GUI, the first operation is to select one of the common ports (COM PORT) to link the portable signal acquisition device, as shown in Figure 6. Once this connection has been made, the Connect button on the display is indented. If the portable signal acquisition device and the NB computer have been successfully linked, the Connect button will change to Disconnect, the other commands such as the ADC ON, Load Flash and Clear Flash will appear, and any one of them can be used. After the ADC ON button has been pressed, the measured signals will be immediately sent to the NB computer. Then, users can choose which channels will be used to display the signals, as shown in Figure 6. There are three kinds of measured signals respectively from the channels 1, 2 and 4 that can be displayed on the screen, ECG, PPG and heart sound signals. To stop the ADC operation, press the ADC OFF button. If users want to record the signals, they can press the Write button. These signals will be recorded on the flash memory in real time. For each new recording event, the file number in the flash memory is automatically increased by one, and the data will be recorded on a new page of the flash memory.
5) The authors have not mentioned in the manuscript that their system is capable of which physiological sensors to acquire the data.
Ans: We proposed a portable multi-channel acquisition system in this study. We did not develop the sensors modules. The analog signals detected by the sensor modules (commercial products) all can be recorded by our device. We have explained this issue in Discussion Section.
In this study, a portable and wireless multi-channel acquisition system for the physiological signal measurement has been fully established. This acquisition system primarily comprises a portable acquisition device, a GUI and an APP to display the signals on a NB computer or a smart device. Three major advantages exist in the present multi-channel acquisition system. First, this acquisition system allows users to combine it with a NB computer or a smart device to display the measured signals in real time and to easily control the functions of the portable acquisition device. Therefore, users are permitted to check the stability and accuracy of the recorded signals in real time during the experiment. When the correctness of the measured signals is carefully confirmed, users can begin to record the signals on the flash memory. Second, the portable signal acquisition device possesses superior mobility. Users may take the device to any place due to its own independent power system and adequate memory. Third, the acquisition system has good expandability. For instance, it offers ± 3.3 voltage for the usage of external or additional circuits. Furthermore, with eight channels, it can be applied to simultaneously register various physiological signals. Because the resolution of ADC is 80.6 uV, it could be used to properly measure several kinds of physiological signals, like as ECG, EMG, PPG, body acceleration signal, and so on.
6) In the proposed system, the authors are using the USB port for charging the battery and also for data movement to the NB (Figure 3 (a)). But authors have not mentioned that whether they have multiple USB ports in the system or they have used a single USB port with the time multiplexing capability.
Ans: A single USB port can be used to charge the battery, or to transmit data to NB computer. We have described this point in 4.1 Section.
4.1. Circuit of Portable Signal Acquisition Device
Figure 5(a) shows the block diagram of the portable acquisition device, Figure 5(b) shows its printed circuit, and Figure 5(c) shows its real photograph. Without the battery, its weight is only 15 g, and its size is 12 cm x 12 cm. In the upper right-hand corner, two voltage sources (3.3 V and -3.3 V) provide the power for external sensor devices with two maximum currents of 400 mA and 60 mA, respectively. The USB port is used either to charge the lithium battery or to download the data to the hard disk in the NB computer. The USB socket is placed at the upper left- hand corner of the printed circuit board. The MCU, BT module, and flash memory consume the largest amount of power in this portable acquisition device. Since the active current of the MSP430 F5438A is 165 μA/MHz, it requires the currents of 3.96 mA and 2.6 μA under the maximum running condition and the low power mode, respectively. The BT model needs the currents of 37 mA and 70 μA under the transmission and standby modes, respectively. Also, the flash memory needs the currents of 40 mA and 70 μA under the operation and standby modes, respectively. Therefore, when the portable acquisition device makes use of a smart device to display the signals, it requires a maximum current of about 43 mA. But, if users want to display signals and write data to the flash memory at the same time, the maximum consumed current will be 73 mA.
7) In the proposed system, the authors are sending the data to the NB by default to display on the GUI then why are they using the onboard flash memory whereas they can easily store the data in the NB.
Ans: We have explained the reason in Discussion Section.
The commercial products, like as the BioPac Bionomadix smart center [3] and BioradioTM wireless measurement system [4], have the exclusive sensor modules and must connect with the NB. These measurement systems just like as gateways to transmit the signals to NB computer. Therefore, their compatibilities are not higher than our proposed system. Users need to carry a NB computer to record these signals during the experimental course. For our system, users only use a smart phone to check the measuring signals and to see whether they are stable and accurate or not when doing the experiment. Once the signals are properly measured, users can control the portable acquisition device to record the signals onto the flash memory in real time. Moreover, since our proposed system is intended to be employed for signal acquisition, it does not need to do the calibration of the sensor modules.
8) In the manuscript the authors have not verified the acquired signals using the statistical data.
Ans: Because our proposed acquisition system basically belongs to a digital one, we may be not necessary to make any calibration of the digital signal or statistical analysis for the recorded data. We have explained this in Discussion Section.
In this study, a portable and wireless multi-channel acquisition system for the physiological signal measurement has been fully established. This acquisition system primarily comprises a portable acquisition device, a GUI and an APP to display the signals on a NB computer or a smart device. Three major advantages exist in the present multi-channel acquisition system. First, this acquisition system allows users to combine it with a NB computer or a smart device to display the measured signals in real time and to easily control the functions of the portable acquisition device. Therefore, users are permitted to check the stability and accuracy of the recorded signals in real time during the experiment. When the correctness of the measured signals is carefully confirmed, users can begin to record the signals on the flash memory. Second, the portable signal acquisition device possesses superior mobility. Users may take the device to any place due to its own independent power system and adequate memory. Third, the acquisition system has good expandability. For instance, it offers ± 3.3 voltage for the usage of external or additional circuits. Furthermore, with eight channels, it can be applied to simultaneously register various physiological signals. Because the resolution of ADC is 80.6 uV, it could be used to properly measure several kinds of physiological signals, like as ECG, EMG, PPG, body acceleration signal, and so on.
The commercial products, like as the BioPac Bionomadix smart center [3] and BioradioTM wireless measurement system [4], have the exclusive sensor modules and must connect with the NB. These measurement systems just like as gateways to transmit the signals to NB computer. Therefore, their compatibilities are not higher than our proposed system. Users need to carry a NB computer to record these signals during the experimental course. For our system, users only use a smart phone to check the measuring signals and to see whether they are stable and accurate or not when doing the experiment. Once the signals are properly measured, users can control the portable acquisition device to record the signals onto the flash memory in real time. Moreover, since our proposed system is intended to be employed for signal acquisition, it does not need to do the calibration of the sensor modules.
9) In this manuscript the authors have not made any discussion about the signal o Noise Ratio for their developed system.
Ans: Since our proposed acquisition system basically belongs to a digital one, it is not necessary to make any calibration of the digital signal or statistical analysis for the recorded data. We have explained this in Discussion Section.
10) The authors have mentioned about any precautionary or security measure for the communication errors or data loss while transmitting data to and from the developed system.
Ans: Because the BT communication technique will verify the correctness of data transmission, we are not necessary to do the acknowledgment for the data transmission.
11) The authors have not mentioned any LED numbers for the specified jobs which they have mentioned in the line numbers 201 and 202.
Ans: We have mentioned LED functions in 3.4 Section, and the LED numbers are shown in Figure 5(c).
3.4. The Setting of MSP430F5438A
The MSP430 F5438A, a 16-bit MCU, used in this study has one flash memory of 256k bytes and eight ADCs of 12-bit. Each ADC channel with a sampling frequency of 500 HZ and an internal flash memory are used to record the number and the size of the files accessed in the external flash memory. The firmware flowchart for the MSP430 microcontroller is shown in Figure 4. There are two UART sets in the 16-bit MCU. One is used for the BT module and the other for the USB port. When the measured signals are sent to a smart device to determine whether they are saved to the flash memory or not, the sampling frequency is reduced to 100 Hz. The main clock rate of the MCU is set at 24 MHz. Clock A0, at 500 Hz, is employed as the ADC sampling frequency and to set up the UART and SPI clocks. The eight channels are made available for the ADC operation. After initial settings are completed, time interruption is enabled and the MCU will wait for a command, such as an “ADC on”, a “Write data” to flash memory, and so on.
The MCU determines whether the flash memory has enough free space based on two conditions: 1) there are 682 files in the flash memory, and 2) the last page of the flash memory has been used. When either of these two conditions is satisfied, the data can no longer be written to the flash memory.
The MCU controls a set of four LEDs indicating: 1) the status of the connection to the USB port, 2) the status of the BT connection, 3) the status of the 3.3 V power, and 4) the status of the flash memory, respectively. If the MCU is in the reading mode, the LED will remain on. However, when in the writing mode, a blinking LED is displayed.
(c)
Figure 5. The portable acquisition device, (a) its block diagram, (b) its printed circuit, (c) its real photo.
12) In Discussion section the authors should discuss about the results they have obtained, any flaws which they have in system and/or the future of the developed system. There are some spelling and indentation errors in the manuscript, which need to be corrected.
Ans: We have discussed our results in Discussion Section and described the future work in Conclusion Section.
5. DiscussionIn the last decade, wearable devices have been applied in various fields, especially in the healthcare area [17]. Among the physiological measurements, the ECG, EMG, EEG and PPG signals, are the most popular in modern wearable devices for daily healthcare monitoring. However, one of the challenges with designing such wearable devices is how to faithfully acquire these physiological signals under an actual environment. Requirements for the wearable healthcare devices are somewhat different from those for the traditional devices in clinical locations. A big difference is that a wearable device always not only detects the dynamic physiological signals, but also needs to display these signals in real time. In recent studies, some acquisition systems have employed the BT, Zigbee, WiFi or RFID technology to transmit the measured signals to a NB computer or a smart device, and to display or store them on those smart devices [5-8]. Nevertheless, there are two main disadvantages in those acquisition systems, including low mobility and low sampling rates.
In this study, a portable and wireless multi-channel acquisition system for the physiological signal measurement has been fully established. This acquisition system primarily comprises a portable acquisition device, a GUI and an APP to display the signals on a NB computer or a smart device. Three major advantages exist in the present multi-channel acquisition system. First, this acquisition system allows users to combine it with a NB computer or a smart device to display the measured signals in real time and to easily control the functions of the portable acquisition device. Therefore, users are permitted to check the stability and accuracy of the recorded signals in real time during the experiment. When the correctness of the measured signals is carefully confirmed, users can begin to record the signals on the flash memory. Second, the portable signal acquisition device possesses superior mobility. Users may take the device to any place due to its own independent power system and adequate memory. Third, the acquisition system has good expandability. For instance, it offers ± 3.3 voltage for the usage of external or additional circuits. Furthermore, with eight channels, it can be applied to simultaneously register various physiological signals. Because the resolution of ADC is 80.6 uV, it could be used to properly measure several kinds of physiological signals, like as ECG, EMG, PPG, body acceleration signal, and so on.
The commercial products, like as the BioPac Bionomadix smart center [3] and BioradioTM wireless measurement system [4], have the exclusive sensor modules and must connect with the NB. These measurement systems just like as gateways to transmit the signals to NB computer. Therefore, their compatibilities are not higher than our proposed system. Users need to carry a NB computer to record these signals during the experimental course. For our system, users only use a smart phone to check the measuring signals and to see whether they are stable and accurate or not when doing the experiment. Once the signals are properly measured, users can control the portable acquisition device to record the signals onto the flash memory in real time. Moreover, since our proposed system is intended to be employed for signal acquisition, it does not need to do the calibration of the sensor modules.
6. ConclusionsIn summary, we designed the portable and wireless multi-channel acquisition system which offered a convenient and portable studying tool to measure the dynamic physiological signals in healthcare research fields. Users can watch the measured signals in real time and control the functions of this multi-channel acquisition device with a smart phone or NB computer. Also, researchers can design their own sensing circuits and combine them with this acquisition system. Then, they can rapidly perform their experiments and with an easy mind collect physiological signals of interest under a ubiquitous scenario. In this study, our proposed acquisition system is currently in a prototype status. However, in the future, we will modify this system as a commercial product according to the feedback comments of the intended audiences.
13) There are some spelling and indentation errors in the manuscript, which need to be corrected.
Ans: We have revised this manuscript again and correct these mistakes.

Reviewer 2 Report
The paper deals with the development of a portable datalogger, which can be operated with a notebook or a tablet. The device can measure up to 8 channels, a low power MCU, Bluetooth 3.0 and 2 GB on-board. Signals can be displayed real-time. The paper discusses the background, the power performance, the circuit, etc.
Scientifically the paper hardly provides new ideas or topics beyond state-of-the art, or it is not described in the text clearly. There are commercial portable devices available. The authors should carefully revisit the content to clarify the main new items for their system. The text also requires improvements in writing.
More specific comments on content:
The requirements for the system design should be discussed in detail and how this influenced the development. A mobile portable device requires a careful housing design / user interface for acceptance by the intended users. This last point is not well covered. There is no section to discuss the final system e.g. with a picture to get an impression of handling aspects. What is the size of the device? What is the weight? etc Did the author follow applicable technical standards for the system design? Has the system being tested by the intended audience to get feedback on acceptance and improvements?Author Response
To Reviewer #2:
Thank the second reviewer for his/her valuable comments that better this manuscript. The texts in the revised manuscript have been corrected/ modified by yellow marker.
Comments to the Author
The paper deals with the development of a portable datalogger, which can be operated with a notebook or a tablet. The device can measure up to 8 channels, a low power MCU, Bluetooth 3.0 and 2 GB on-board. Signals can be displayed real-time. The paper discusses the background, the power performance, the circuit, etc.
1) Scientifically the paper hardly provides new ideas or topics beyond state-of-the art, or it is not described in the text clearly. There are commercial portable devices available. The authors should carefully revisit the content to clarify the main new items for their system.
Ans.: We have modified the Introduction section and reviewed some commercial products which are similar to our proposed system.
IntroductionMost of the fitness and healthcare devices are capable of dynamic monitoring, such as pedometers and heart rate monitors. These wearable devices must have the function of an independent power supply and the ability of wireless transmission. Typically, they measure and deal with these physiological signals by means of a microcontroller unit (MCU) or send these signals to a tablet or a smart phone to further process. Therefore, in the development of wearable healthcare devices, it is important to explore innovative measurement technologies and signal processing methods to overcome the instability relevant to the acquisition of dynamic physiological signals. For these researchers in the hardware development, they must not only develop novel analog driving circuits, but also build a standard acquiring system which is supposed to incorporate a power system, an analog-to-digital converter and a wireless transmission scheme. Moreover, they also need to build a display and control interface in the software development. It is a really time- and energy-consuming task for those researchers in their ordinary studying procedure. Thus, how to efficiently shorten their research time may become a serious issue.
Multi-channel physiological signal measurement systems that are employed in a lot of physiological experiments usually require an alternating current (AC) and must connect the different sensor modules with the lead wires. For example, both the BioPac MP150 system and the K&H KL-710 system can connect with specific physiological measurement modules and then record and display multiple physiological signals on a notebook (NB) or personal computer [1,2]. Obviously, several researchers make use of compact DAQ cards to assemble their own signal acquisition systems, which connect with NB computers through a universal serial bus (USB) cable. However, the dimension of these acquisition systems would be huge, and their powers are supported by an AC or by a built-in battery in the NB computer. These two acquisition systems, the BioPac MP150 system and the K&H KL-710 system, neither have an independent data storage unit, nor have a wireless transmission function. Moreover, these two acquisition systems are not able to be operated by an application program (APP) using a tablet or a smart phone. Due to those serious limitations, it is difficult for researchers to apply these systems in developing their own wearable devices. In addition, the BioPac Bionomadix smart center can connect three wireless dual-channel modules [3], and the BioradioTM, with eight channels, is a wireless measurement system which can concurrently measure six different physiological signals [4].
Since Bluetooth (BT) technology was not prevalent in the iOS and the Android operating systems ten years ago, and the performance of the central processing unit (CPU) in mobile phones was relatively low, a mobile phone did not was used to concurrently display and deal with these measured signals in real time. The BT3.0 and BT4.0 dual-channel modules both have become the standard components of smart phones for the past few years, and the CPU performance in modern mobile phones has been considerably upgraded. Therefore, it is now feasible to develop a portable wireless system based on the traditional NB computer or smart phone that can measure, store and exhibit multiple physiological signals in real time. This acquired system can really help researchers to develop a wearable device to apply in the fitness and healthcare fields.
In previous studies, Lin et al. [5] adopted the MSP430 microcontroller as the core of the IoT based wireless polysomnography intelligent system. Their system can transmit the measured signals directly to a mobile phone for storage and analysis. Because the problem of the slow BT transmission speed, the sampling frequency of the system had to be reduced to 125 Hz, and the system lacked the function to access data. Some studies used the wireless technique, the IEEE 802.15.4 standard, to transmit physiological signals to a personal computer. Biagetti et al. built an acquisition system to measure the electrocardiogram (ECG) and electromyogram (EMG) signals by applying a RF transceiver for wireless communication [6]. Of course, each wireless note also needs the power to transmit data to the server station. Bhutta et al. used the WiFi technique to transmit data [7]. Dey et al. utilized the Zigbee technology to establish a wireless sensor network for the ECG measurements [8]. Apparently, all these studies do not connect directly with smart devices, such as a tablet or a smart phone. Therefore, these studies may be limited to doing the experiments under ubiquitous situations. Liu et al. developed a wearable device that could measure the ECG and gestural movement signals [9], but this device could not demonstrate the recorded signals in real time. On the other hand, the data had to be offline downloaded from the device to display the measured signals. Hsu et al. measured the galvanic skin response with the electrical impedance spectroscopy, which used the BT technology to transmit data to smart phones [10]. Milici et al. designed a thermo sensor to detect the changing in airflow during breathing [11]. They also used the BT technique to transmit the acquired data to a smart phone for long-term recording. Meanwhile, Liu et al. have developed a preliminary prototype of a portable multi-channel physiological measurement system [12].
Since many sensors and their driver circuits have been integrated as chips, their power consumption has been reduced and their size has been miniaturized. For example, the Analog Device AD8232 is an analogy integrated chip for the ECG measurement [13], while the MAX30101 of Maxim Integrated is a digital chip for the oxyhemoglobin saturation assessment [14]. The ADXL325 of the Analog Device is an analog tri-axial accelerometer integrated chip applied for the object activity measurement [15]. As the sensor modules are employed to detect a variety of physiological signals, an acquisition system with multiple channels is required to show and record these signals. However, among the current commercial products, merely a few devices not only can be controlled with a tablet or a smart phone, but also can store data on them. Thus, the goal of this study is to develop a portable and wireless multi-channel acquisition system for the physiological signal measurements. It has eight analog channels and can be controlled by a NB computer or a smart device. The measured signals can either be shown on a NB computer or a smart device in real time, or be stored on the flash memory of the portable acquisition device. A sampling frequency of the portable signal acquisition device is 500 Hz which is enough to conform to the Nyquist frequency of some physiological signals, like as the ECG, electroencephalogram (EEG), electrooculogram (EOG), galvanic skin response (GSR), and photoplethysmogram (PPG), since most of them do not have a large bandwidth [16]. With a TI MSP430 F5438A as its MCU, the portable acquisition device has a compact size, uses a lithium battery (350 mA) to supply the needed power, employs a BT3.0 module to transmit data, and a 2GB flash memory to store the measured signals. Moreover, the portable acquisition device can offer dual power levels, ±3 voltage, so that the external sensor modules may connect with this device to measure different physiological signals. The real-time measuring signals can be displayed on a NB computer or smart phone. Thus, consumers or researchers can confirm the stability and accuracy of the measured signals during the experiment.
The rest of the paper is organized as follows: Section 2 describes the structure of the multi-channel acquisition system, and its software commands on both smart devices and NB computers. Section 3 describes the hardware and firmware designs of the portable signal acquisition device. Section 4 presents the results, and the discussion and conclusions are drawn in Section 5 and 6.
2) The text also requires improvements in writing.
Ans.: We have revised this manuscript again and corrected these mistakes.
3) The requirements for the system design should be discussed in detail and how this influenced the development.
Ans.: We have discussed our results in Discussion Section and described the future work in Conclusion Section.
5. DiscussionIn the last decade, wearable devices have been applied in various fields, especially in the healthcare area [17]. Among the physiological measurements, the ECG, EMG, EEG and PPG signals, are the most popular in modern wearable devices for daily healthcare monitoring. However, one of the challenges with designing such wearable devices is how to faithfully acquire these physiological signals under an actual environment. Requirements for the wearable healthcare devices are somewhat different from those for the traditional devices in clinical locations. A big difference is that a wearable device always not only detects the dynamic physiological signals, but also needs to display these signals in real time. In recent studies, some acquisition systems have employed the BT, Zigbee, WiFi or RFID technology to transmit the measured signals to a NB computer or a smart device, and to display or store them on those smart devices [5-8]. Nevertheless, there are two main disadvantages in those acquisition systems, including low mobility and low sampling rates.
In this study, a portable and wireless multi-channel acquisition system for the physiological signal measurement has been fully established. This acquisition system primarily comprises a portable acquisition device, a GUI and an APP to display the signals on a NB computer or a smart device. Three major advantages exist in the present multi-channel acquisition system. First, this acquisition system allows users to combine it with a NB computer or a smart device to display the measured signals in real time and to easily control the functions of the portable acquisition device. Therefore, users are permitted to check the stability and accuracy of the recorded signals in real time during the experiment. When the correctness of the measured signals is carefully confirmed, users can begin to record the signals on the flash memory. Second, the portable signal acquisition device possesses superior mobility. Users may take the device to any place due to its own independent power system and adequate memory. Third, the acquisition system has good expandability. For instance, it offers ± 3.3 voltage for the usage of external or additional circuits. Furthermore, with eight channels, it can be applied to simultaneously register various physiological signals. Because the resolution of ADC is 80.6 uV, it could be used to properly measure several kinds of physiological signals, like as ECG, EMG, PPG, body acceleration signal, and so on.
The commercial products, like as the BioPac Bionomadix smart center [3] and BioradioTM wireless measurement system [4], have the exclusive sensor modules and must connect with the NB. These measurement systems just like as gateways to transmit the signals to NB computer. Therefore, their compatibilities are not higher than our proposed system. Users need to carry a NB computer to record these signals during the experimental course. For our system, users only use a smart phone to check the measuring signals and to see whether they are stable and accurate or not when doing the experiment. Once the signals are properly measured, users can control the portable acquisition device to record the signals onto the flash memory in real time. Moreover, since our proposed system is intended to be employed for signal acquisition, it does not need to do the calibration of the sensor modules.
6. Conclusions
In summary, we designed the portable and wireless multi-channel acquisition system which offered a convenient and portable studying tool to measure the dynamic physiological signals in healthcare research fields. Users can watch the measured signals in real time and control the functions of this multi-channel acquisition device with a smart phone or NB computer. Also, researchers can design their own sensing circuits and combine them with this acquisition system. Then, they can rapidly perform their experiments and with an easy mind collect physiological signals of interest under a ubiquitous scenario. In this study, our proposed acquisition system is currently in a prototype status. However, in the future, we will modify this system as a commercial product according to the feedback comments of the intended audiences.
4) A mobile portable device requires a careful housing design / user interface for acceptance by the intended users. This last point is not well covered.
Ans.: In this study, we only propose a prototype of the portable acquisition system. Researchers may only make their own sensor modules and analog driving circuits, and then they can start their healthcare-related experiments. In the future, we will modify this system according to the comments of intended users. We have described this in Conclusion Section.
6. ConclusionsIn summary, we designed the portable and wireless multi-channel acquisition system which offered a convenient and portable studying tool to measure the dynamic physiological signals in healthcare research fields. Users can watch the measured signals in real time and control the functions of this multi-channel acquisition device with a smart phone or NB computer. Also, researchers can design their own sensing circuits and combine them with this acquisition system. Then, they can rapidly perform their experiments and with an easy mind collect physiological signals of interest under a ubiquitous scenario. In this study, our proposed acquisition system is currently in a prototype status. However, in the future, we will modify this system as a commercial product according to the feedback comments of the intended audiences.
5) There is no section to discuss the final system e.g. with a picture to get an impression of handling aspects. What is the size of the device? What is the weight? etc Did the author follow applicable technical standards for the system design?
Ans.: We have described the portable acquisition device in 4.1 Section, and discussed our proposed system in Discussion Section.
4.1. Circuit of Portable Signal Acquisition Device
Figure 5(a) shows the block diagram of the portable acquisition device, Figure 5(b) shows its printed circuit, and Figure 5(c) shows its real photograph. Without the battery, its weight is only 15 g, and its size is 12 cm x 12 cm. In the upper right-hand corner, two voltage sources (3.3 V and -3.3 V) provide the power for external sensor devices with two maximum currents of 400 mA and 60 mA, respectively. The USB port is used either to charge the lithium battery or to download the data to the hard disk in the NB computer. The USB socket is placed at the upper left- hand corner of the printed circuit board. The MCU, BT module, and flash memory consume the largest amount of power in this portable acquisition device. Since the active current of the MSP430 F5438A is 165 μA/MHz, it requires the currents of 3.96 mA and 2.6 μA under the maximum running condition and the low power mode, respectively. The BT model needs the currents of 37 mA and 70 μA under the transmission and standby modes, respectively. Also, the flash memory needs the currents of 40 mA and 70 μA under the operation and standby modes, respectively. Therefore, when the portable acquisition device makes use of a smart device to display the signals, it requires a maximum current of about 43 mA. But, if users want to display signals and write data to the flash memory at the same time, the maximum consumed current will be 73 mA.
(a)
(b)
(c)
Figure 5. The portable acquisition device, (a) its block diagram, (b) its printed circuit, (c) its real photograph.
5. Discussion
In the last decade, wearable devices have been applied in various fields, especially in the healthcare area [17]. Among the physiological measurements, the ECG, EMG, EEG and PPG signals, are the most popular in modern wearable devices for daily healthcare monitoring. However, one of the challenges with designing such wearable devices is how to faithfully acquire these physiological signals under an actual environment. Requirements for the wearable healthcare devices are somewhat different from those for the traditional devices in clinical locations. A big difference is that a wearable device always not only detects the dynamic physiological signals, but also needs to display these signals in real time. In recent studies, some acquisition systems have employed the BT, Zigbee, WiFi or RFID technology to transmit the measured signals to a NB computer or a smart device, and to display or store them on those smart devices [5-8]. Nevertheless, there are two main disadvantages in those acquisition systems, including low mobility and low sampling rates.
In this study, a portable and wireless multi-channel acquisition system for the physiological signal measurement has been fully established. This acquisition system primarily comprises a portable acquisition device, a GUI and an APP to display the signals on a NB computer or a smart device. Three major advantages exist in the present multi-channel acquisition system. First, this acquisition system allows users to combine it with a NB computer or a smart device to display the measured signals in real time and to easily control the functions of the portable acquisition device. Therefore, users are permitted to check the stability and accuracy of the recorded signals in real time during the experiment. When the correctness of the measured signals is carefully confirmed, users can begin to record the signals on the flash memory. Second, the portable signal acquisition device possesses superior mobility. Users may take the device to any place due to its own independent power system and adequate memory. Third, the acquisition system has good expandability. For instance, it offers ± 3.3 voltage for the usage of external or additional circuits. Furthermore, with eight channels, it can be applied to simultaneously register various physiological signals. Because the resolution of ADC is 80.6 uV, it could be used to properly measure several kinds of physiological signals, like as ECG, EMG, PPG, body acceleration signal, and so on.
The commercial products, like as the BioPac Bionomadix smart center [3] and BioradioTM wireless measurement system [4], have the exclusive sensor modules and must connect with the NB. These measurement systems just like as gateways to transmit the signals to NB computer. Therefore, their compatibilities are not higher than our proposed system. Users need to carry a NB computer to record these signals during the experimental course. For our system, users only use a smart phone to check the measuring signals and to see whether they are stable and accurate or not when doing the experiment. Once the signals are properly measured, users can control the portable acquisition device to record the signals onto the flash memory in real time. Moreover, since our proposed system is intended to be employed for signal acquisition, it does not need to do the calibration of the sensor modules.
6) Has the system being tested by the intended audience to get feedback on acceptance and improvements?
Ans.: In this study, we only propose a prototype of the portable acquisition system. Researchers may only make their own sensor modules and analog driving circuits, and then they can start their healthcare-related experiments. In the future, we will modify this system according to the comments of intended users. We have mentioned this point in Conclusion Section.

Round 2
Reviewer 1 Report
The reviewer has found significant improvement in the revised manuscript as compared to the first version, but there are still some uncertainties which need to be answered before the manuscript gets accepted for publication.
In the first revision the reviewer has asked that describe the reason of using Bluetooth technology over the WiFi technology but the authors have not included the comparison of two technologies and why Bluetooth is more suitable for their design. The authors have mentioned in the revised manuscript that either they are using the single USB port for charging the battery and data movement but they have mentioned that they are doing both jobs simultaneously or these processes are done at different times. If they are getting done simultaneously then how the authors are able to do that and if they are separate, then based on which control signal the functionality of the USB port is divided. The authors have not used any statistical method to verify the data and also have not mentioned anything about the signal to noise ratio (SNR) in the revised manuscript. The authors have only mentioned that their system is a digital system so they do not need to verify the data and no need to mention about the SNR. Even the data from the microcontroller is digital but the authors are receiving the analog signals from the sensors and then converting them to digital using the ADC in the microcontroller so the verification of the signals is needed to justify the correct conversion and transmission of the signals.Author Response
To Reviewer #1:
Thank the first reviewer for his/her valuable comments that make better this manuscript. The texts in this revised manuscript have been corrected/ modified by red words.
Comments to the Author
The first revision the reviewer has asked that describe the reason of using Bluetooth technology over the WiFi technology but the authors have not included the comparison of two technologies and why Bluetooth is more suitable for their design.
Ans: The authors have added some paragraphs to compare with the four wireless techniques in Discussion section.
In the last decade, wearable devices have been applied in various fields, especially in the healthcare area [17]. Among the physiological measurements, the ECG, EMG, EEG and PPG signals, are the most popular in modern wearable devices for daily healthcare monitoring. However, one of the challenges with designing such wearable devices is how to faithfully acquire these physiological signals under an actual environment. Requirements for the wearable healthcare devices are somewhat different from those for the traditional devices in clinical locations. A big difference is that a wearable device always not only detects the dynamic physiological signals, but also needs to display these signals in real time. In recent studies, some acquisition systems have employed the BT, Zigbee, WiFi or RF transceiver technology to transmit the measured signals to a NB computer or a smart device, and to display or store them on those smart devices [5-8]. The characteristic comparison among the four wireless techniques are shown in Table 3. Although the WiFi has the fastest transmission rate and the longest transmission distance, its power consumption also is the largest. The BT does not have the best performance in these wireless techniques. But, nowadays, both the NB computer and smart phone all have the WiFi and BT communication function. Nevertheless, there are two main disadvantages in those acquisition systems, including low mobility and low sampling rates.
Table 3. The characteristic comparison among the four wireless techniques.
|
Technique |
Distance |
Power consumption |
Transmission rate |
|
BT 3.0 (HC-05) [18] |
< 10 m |
30 mA |
1M (bits/s) |
|
WiFi (ESP8285) [19] |
> 10 m |
80 mA |
54M (bits/s) |
|
Zigbee (Xbee®) [20] |
> 10 m |
45 mA |
250k (bits/s) |
|
RF (BC9824) [21] |
< 2 m |
10 mA |
2M (bits/s) |
In this study, a portable and wireless multi-channel acquisition system for the physiological signal measurement has been fully established. This acquisition system primarily comprises a portable acquisition device, a GUI and an APP to display the signals on a NB computer or a smart device. Since the power consumption of the BT technique is relatively low as compared with WiFi, we finally choose the BT technique to perform the connection with the NB computer or the smart device in order to reduce the total power consumption of the portable acquisition device. Three major advantages exist in the present multi-channel acquisition system. First, this acquisition system allows users to combine it with a NB computer or a smart device to display the measured signals in real time and to easily control the functions of the portable acquisition device. Therefore, users are permitted to check the stability and accuracy of the recorded signals in real time during the experiment. When the correctness of the measured signals is carefully confirmed, users can begin to record the signals on the flash memory. Second, the portable signal acquisition device possesses superior mobility. Users may take the device to any place due to its own independent power system and adequate memory. Third, the acquisition system has good expandability. For instance, it offers ± 3.3 voltage for the usage of external or additional circuits. Furthermore, with eight channels, it can be applied to simultaneously register various physiological signals. Because the resolution of ADC is 80.6 uV, it could be used to properly measure several kinds of physiological signals, like as ECG, EMG, PPG, body acceleration signal, and so on.
Authors have mentioned in the revised manuscript that either they are using the single USB port for charging the battery and data movement but they have mentioned that they are doing both jobs simultaneously or these processes are done at different times. If they are getting done simultaneously then how the authors are able to do that and if they are separate, then based on which control signal the functionality of the USB port is divided.
Ans: We have described how to use the USB port to supply the power source and transmit data in our portable acquired device in 4.1 section.
4.1. Circuit of Portable Signal Acquisition Device
Figure 5(a) shows the block diagram of the portable acquisition device, Figure 5(b) shows its printed circuit, and Figure 5(c) shows its real photograph. Without the battery, its weight is only 15 g, and its size is 12 cm x 12 cm. In the upper right-hand corner, two voltage sources (3.3 V and -3.3 V) provide the power for external sensor devices with two maximum currents of 400 mA and 60 mA, respectively. The USB port is used either to charge the lithium battery or to download the data to the hard disk in the NB computer. In Figure 5(a), the black lines represent the power connection, and the blue lines represent the data transmission. The 5V line and ground line of the USB port were connected to the input pins of the charge IC, TI BQ24072. The two data lines of the USB port were connected to the USB-to-Serial Bridge Controller, PL 2303HX. The USB socket is placed at the upper left- hand corner of the printed circuit board. The MCU, BT module, and flash memory consume the largest amount of power in this portable acquisition device. Since the active current of the MSP430 F5438A is 165 μA/MHz, it requires the currents of 3.96 mA and 2.6 μA under the maximum running condition and the low power mode, respectively. The BT model needs the currents of 37 mA and 70 μA under the transmission and standby modes, respectively. Also, the flash memory needs the currents of 40 mA and 70 μA under the operation and standby modes, respectively. Therefore, when the portable acquisition device makes use of a smart device to display the signals, it requires a maximum current of about 43 mA. But, if users want to display signals and write data to the flash memory at the same time, the maximum consumed current will be 73 mA. When the input of ADC is connected to the power source, +3.3 V, and ground, 0 V, the codes of ADC with the ten samples are shown in Table 2. The statistical codes for the positive power source and ground source are 4095 ± 1 (mean ± standard deviation) and 1 ± 1, respectively.
(a)
Authors have not used any statistical method to verify the data and also have not mentioned anything about the signal to noise ratio (SNR) in the revised manuscript. The authors have only mentioned that their system is a digital system so they do not need to verify the data and no need to mention about the SNR. Even the data from the microcontroller is digital but the authors are receiving the analog signals from the sensors and then converting them to digital using the ADC in the microcontroller so the verification of the signals is needed to justify the correct conversion and transmission of the signals.
Ans: We have added the statistical analysis for the ADC function in 4.1 section.
4.1. Circuit of Portable Signal Acquisition Device
Figure 5(a) shows the block diagram of the portable acquisition device, Figure 5(b) shows its printed circuit, and Figure 5(c) shows its real photograph. Without the battery, its weight is only 15 g, and its size is 12 cm x 12 cm. In the upper right-hand corner, two voltage sources (3.3 V and -3.3 V) provide the power for external sensor devices with two maximum currents of 400 mA and 60 mA, respectively. The USB port is used either to charge the lithium battery or to download the data to the hard disk in the NB computer. In Figure 5(a), the black lines represent the power connection, and the blue lines represent the data transmission. The 5V line and ground line of the USB port were connected to the input pins of the charge IC, TI BQ24072. The 2 data lines of the USB port were connected to the USB-to-Serial Bridge Controller, PL 2303HX. The USB socket is placed at the upper left- hand corner of the printed circuit board. The MCU, BT module, and flash memory consume the largest amount of power in this portable acquisition device. Since the active current of the MSP430 F5438A is 165 μA/MHz, it requires the currents of 3.96 mA and 2.6 μA under the maximum running condition and the low power mode, respectively. The BT model needs the currents of 37 mA and 70 μA under the transmission and standby modes, respectively. Also, the flash memory needs the currents of 40 mA and 70 μA under the operation and standby modes, respectively. Therefore, when the portable acquisition device makes use of a smart device to display the signals, it requires a maximum current of about 43 mA. But, if users want to display signals and write data to the flash memory at the same time, the maximum consumed current will be 73 mA. When the input of ADC is connected to the power source, +3.3 V, and ground, 0 V, the codes of ADC with the ten samples are shown in Table 2. The statistical codes for the positive power source and ground source are 4095 ± 1 (mean ± standard deviation) and 1 ± 1, respectively.
Table 2. The statistical error of ADC codes under +3.3 V and 0 V.
|
Source |
1 |
2 |
3 |
4 |
5 |
6 |
7 |
8 |
9 |
10 |
Mean ± SD |
|
+3.3 V |
4095 |
4095 |
4095 |
4093 |
4094 |
4095 |
4095 |
4095 |
4094 |
4095 |
4094.5 ± 0.70 |
|
0 V |
1 |
1 |
0 |
1 |
2 |
0 |
0 |
0 |
1 |
0 |
0.6 ± 0.70 |
SD: standard deviation.
